# The impact of school closure intervention during the third wave of the COVID-19 pandemic in Italy: Evidence from the Milan area

**David Consolazio**[1,2], **Simone Sarti**[3], **Marco Terraneo**[2], **Corrado Celata**[4], **Antonio Giampiero Russo**[1] *

**1** Epidemiology Unit, Agency for Health Protection of Milan, Milano, Italy, **2** Department of Sociology and Social Research, University of Milan-Bicocca, Milano, Italy, **3** Department of Social and Political Sciences, University of Milan, Milano, Italy, **4** Specific Prevention Unit, Agency for Health Protection of Milan, Milano, Italy

\* agrusso@ats-milano.it

## Abstract

### Background

In February 2021, the spread of a new variant of SARS-CoV-2 in the Lombardy Region, Italy caused concerns about school-aged children as a source of contagion, leading local authorities to adopt an extraordinary school closure measure. This generated a debate about the usefulness of such an intervention in light of the trade-off between its related benefits and costs (e.g. delays in educational attainment, impact on children and families' psycho-physical well-being). This article analyses the epidemiological impact of the school closure intervention in the Milan metropolitan area.

### Methods

Data from the Agency for Health Protection of the Metropolitan City of Milan allowed analysing the trend of contagion in different age classes before and after the intervention, adopting an interrupted times series design, providing a quasi-experimental counterfactual scenario. Segmented Poisson regression models of daily incident cases were performed separately for the 3–11-year-old, the 12–19-year-old, and the 20+-year-old age groups, examining the change in the contagion curves after the intervention, adjusting for time-varying confounders. Kaplan-Meier survival curves and Cox regression were used to assess the equality of survival curves in the three age groups before and after the intervention.

### Results

Net of time-varying confounders, the intervention produced a daily reduction of the risk of contagion by 4% in those aged 3–11 and 12–19 (IRR = 0·96) and by 3% in those aged 20 or more (IRR = 0·97). More importantly, there were differences in the temporal order of contagion decrease between the age groups, with the epidemic curve lowering first in the school-

**Data Availability Statement:** All relevant data are within the manuscript and its Supporting Information files.

**Funding:** The author(s) received no specific funding for this work.

**Competing interests:** The authors have declared that no competing interests exist.

aged children directly affected by the intervention, and only subsequently in the adult population, which presumably indirectly benefitted from the reduction of contagion among children.

## Conclusion

Though it was not possible to completely discern the effect of school closures from concurrent policy measures, a substantial decrease in the contagion curves was clearly detected after the intervention. The extent to which the slowdown of infections counterbalanced the social costs of the policy remains unclear.

## Introduction

In February 2021, a warning arose in the Lombardy Region, Italy, when four outbreaks of a new variant of SARS-CoV-2 (lineage B·1·1·7) were identified in some schools in the municipalities of Bollate (province of Milan), Castrezzato (Brescia), Mede (Pavia), and Viggiù (Varese). Recent literature warned that the so-called Alpha variant was more contagious than previous ones, with an easier spread among children [1, 2]. In the four municipalities, the increasing number of infected people led to exceeding the alarm threshold established by the Ministry of Health of 250 cases per 100,000 inhabitants, with the consequent implementation of restrictions on mobility and closures of public services (lockdown provisions) from February 17. Together with these restrictions, local authorities ordered the closure of schools of each level of education in the municipalities where the outbreaks took place, declaring them a 'red zone'–previously, kindergartens (3–5 years old), primary schools (6–10 years old), and the first level of secondary school (11 years old) have always remained open in Lombardy, starting from September 2020. In the following days, a widespread increase in the number of infected people was observed in the whole region, leading local authorities to impose a total school closure in Lombardy from March 5, which lasted until April 5.

School closure—with educational activities carried out through distance learning—implies important social costs; thus, it is crucial to understand the efficacy of such interventions considering the trade-off between the flattening of the epidemic curve and the longstanding consequences on children's education and well-being. Missing out on formal learning could have an impact on children's mental health, likely reducing self-control, social competence, and logical deduction amongst other cognitive abilities [3–5] as well as increasing educational inequalities emphasising different resilience between socioeconomically advantaged and disadvantaged families and contexts [6].

Analysing different countries, several scholars have stressed the role of school closures in the mitigation of the COVID-19 spread [7–9], referring to the spreading role of asymptomatic children and the relational school contexts as a nidus of transmission [10], while others estimated a light effect on the spread of COVID-19, especially if compared to the high social cost of this policy [4, 11–13]. However, most of the studies that have been carried out suffer from several limitations—acknowledged by the authors themselves—which make it difficult to understand the real impact of school closure interventions on the spread of contagion. From a methodological standpoint, disentangling the impact of school closures from the impact of other interventions carried out simultaneously might not be feasible, as it is seldom possible to isolate the effect of one containment measure from the others. Moreover, some variants resulted in being associated with higher spread in specific age classes, which are generally less

exposed to contagion, so that aggregated data might conceal specific trends. Indeed, some scholars raised the alarm about the high contagiousness of the new variant identified first in the United Kingdom in late summer 2020, as ' [i]n September, 2020, this variant represented just one in four new diagnoses of COVID-19, whereas by mid-December, this had increased to almost two thirds of new cases in London' [1].

The analysis of school closure interventions lacks counter factuality, and the impact of this kind of policy continues to be unproved or debated, mainly because of the presence of time-variant confounders that could explain the decrease in contagion after the school closures. In this sense, without an experimental design, it is impossible to test the independent role of school closures in shaping the contagion curve. However, some situations permit evaluating a quasi-experimental model considering two homogeneous contextual conditions which are subject to different health policy interventions.

Since November 6, 2020, the Italian government has adopted a novel system of management for pandemic-related interventions according to four different scenarios ('white', 'yellow', 'orange', and 'red' zones), with gradual regional restrictions corresponding to biweekly $R_t$ values ($< 0·5$, $0·5–1$, $1–1·25$, $> 1·5$), as summarised in Table 1. As regards Lombardy, the school closure intervention on 5 March was imposed in the context of a shift from 'orange' to 'strengthen orange' zone (conceived ad hoc), with background restriction measures left almost untouched. All levels of education were moved to distance learning and only other minor restrictions were introduced, namely the prohibition against (i) using playgrounds in public parks, though recent literature has shown that the risk of outdoor contagion appears to be very low compared to indoor settings [14, 15]; (ii) reaching second homes, which may reasonably have had a limited impact; (iii) visiting friends and relatives without a 'proven' necessity, which needed only to be self-certified with little possibility of control. Bearing in mind that the 'orange' zone before the shift towards the 'strengthen orange' zone lasted only four days, it is nonetheless important to consider the restrictions previously applied, falling under the 'yellow' zone scenario, and how they differed from the ones in the subsequent scenarios. Three changes apply in the shift from a 'yellow' zone to an 'orange' zone scenario, namely: (i) the closure of museums, which were previously already closed on weekends; (ii) the closure of bars and restaurants, which were previously already closed after 6 p.m.; (iii) the prohibition against moving outside the municipality of residence. This last measure, which appears to be the most stringent one, was actually subject to several exceptions due to reasons that needed only to be self-certified (work motivations, health issues and unspecified urgent reasons could be declared) and did not apply for people living in municipalities with less than 5,000 inhabitants, who could move within a radius of 30 km from their residence. Moreover, for a remarkable portion of the population living in the municipality of Milan (~1.4 million inhabitants; ~182 km$^2$; ~40% of the study area population), this did not alter daily mobility habits.

Accordingly, it is clear that other changes are trivial compared to the school closure intervention, which was the only stringent policy applied to a substantial population group (those aged 3–11) with no exceptions. This contributed to shape a 'natural experiment' situation, with dates occurring after the intervention conceivable as having been 'exposed to treatment' and dates before as 'controls'. Indeed, as the specific intervention addressed was almost exclusively oriented towards school closure, this makes it possible to estimate the net effect of the school closure policy on the spread of contagion with little distortion—a situation that had never occurred previously in the context of the current pandemic in Italy.

In light of the framework outlined, an interrupted time series (ITS) study design allowed assessing the possible change in the slope of the contagion curve after school closures with unprecedented accuracy. Indeed—though not eliminating all the possible confounding factors —the design adopted permitted us to estimate the impact of the intervention in the

**Table 1. Summary of the main restrictive measures applied in each scenario, with relative dates of occurrences in the Lombardy region (changes with respect to the previous scenario in bold).**

| 'White' zone | 'Yellow' zone (1–28 February) | 'Orange' zone (1–4 March) | 'Strengthen Orange' zone (5–14 March) | 'Red' zone (15 March–5 April) |
|---|---|---|---|---|
| $R_t < 0.5$ | $0.5 < R_t < 1$ | $1 < R_t < 1.25$ | | $R_t > 1.5$ |
| • Suspension of activities at higher risk | • **Curfew (10 p.m.– 5 a.m.)** | • Curfew (10 p.m.–5 a.m.) | • Curfew (10 p.m.– 5 a.m.) | • Curfew (10 p.m.– 5 a.m.) |
| | • **Closure of bars and restaurants at 6 p.m.** | • **Prohibition against moving outside the municipality of residence (except for 'proven' needs)** | • Prohibition to move outside the municipality of residence | • **Prohibition of unauthorised mobility (also within the municipality of residence)** |
| | • **Closure of shopping malls on the weekend** | • **Closure of bars and restaurants (delivery and take-away only)** | • Closure of bars and restaurants (delivery and take-away only) | • **Closure of shops and retail markets** |
| | • **Closure of movie theatres, theatres** | • **Closure of movie theatres, theatres, and museums** | • Closure of shopping malls on the weekend | • **Closure of hairdressers and beauticians** |
| | • **Closure of sports centres** | • Closure of shopping malls on the weekend | • Closure of movie theatres, theatres, and museums | • Closure of bars and restaurants (delivery and take-away only) |
| | • **Closure of betting places** | • Closure of sports centres | • Closure of sports centres | • Closure of shopping malls on the weekend |
| • Closure of museums on the weekend | • **Prohibition on conferences, conventions, and fairs** | • Closure of betting places | • Closure of betting places | • Closure of movie theatres, theatres, and museums |
| • Local quarantines if needed | • Closure of museums on the weekend | • Prohibition on conferences, conventions, and fairs | • Prohibition on conferences, conventions, and fairs | • Closure of sports centres |
| | • **Distance learning for students from the second level of secondary school onward** | • Distance learning for students from the second level of secondary school onward | • **Prohibition against using recreational areas in public parks** | • Closure of betting places |
| | | | • **Prohibition against reaching second homes** | • Prohibition on conferences, conventions, and fairs |
| | | | • **Prohibition against visiting friends and relatives without a 'proven' necessity** | • Distance learning for students at all school levels |
| | | | • **Distance learning for students at all school levels** | |

Note: The effective reproduction number (*Rt*) represents the average number of new infections caused by a single infected individual at time *t* in the partially susceptible population. *Rt* values below 1 indicate that the epidemic is slowing down (each patient infects, on average, less than 1 person), whereas values above 1 indicate that the epidemic is progressing (each patient infects, on average, more than 1 person).

correspondence of minor concurrent changes. Individual data from the Agency for Health Protection of the Metropolitan City of Milan (ATS of Milan) were aggregated daily to examine the contagion trend in the population directly affected by the school closures (3–11 years old) and in the remaining population who had completed secondary education (20+ years old), relying on Poisson regression models.

## Data and methods

### Study population

We analysed data stemming from the ATS of Milan, covering 193 municipalities belonging to the provinces of Milan and Lodi, in the Lombardy region, with an overall population of nearly 3.48 million people in 2020. The study area includes one of the municipalities at the origin of the Alpha variant outbreak, namely Bollate, which has been subject to specific school closures from February 11 and was declared a 'red' zone on February 17. The study was performed on the whole ATS population excluding Bollate, whose early Alpha variant outbreak and policy interventions may have influenced the wider area's results.

## Measures

From the *Integrated Datawarehouse for COVID Analysis in Milan*, we extracted all subjects with nasopharyngeal swab-confirmed (both molecular and antigenic) SARS-CoV-2 infection from 1 February ('yellow' zone entry date) as of 5 April (last day of the school closures). Data were aggregated daily to examine trends in incident COVID-19 cases. From the same data source, the overall daily number of tested subjects (both positive and negative) was extracted. The daily number of incident cases and the positivity rate (the share of tests returning a positive result) in the country were extracted from the *Italian Civil Protection's* open archive (https://github.com/pcm-dpc/COVID-19). A variable representing the underlying scenario in each date was built relying on the indications available in the national and regional measures to deal with the emergency linked to the spread of the COVID-19 pandemic (https://anci. lombardia.it/dettaglio-news/20202281155-coronavirus-provvedimenti-delle-istituzioni/anci. lombardia.it).

## Statistical analysis

To evaluate the impact of school closures on the spread of contagion, we relied on a segmented regression approach in the context of an ITS study design, which is known to be a valuable tool for assessing the effectiveness of population-level health interventions [16–18]. A randomised controlled trial, which among epidemiological study designs is considered the gold standard for evaluating the effectiveness of an intervention, is not feasible in the context of a public health policy targeted at the population level where a random control group has not been previously identified, and cannot be identified ex-post. However, the ITS study design 'is particularly suited to interventions introduced at a population level over a clearly defined time period and that target population-level health outcomes' [17], and it is often used in the evaluation of 'natural experiments' occurring in real-world settings, where a clear differentiation between the pre- and post-intervention period is present. For our analysis, we followed the methodology proposed by Lopez Bernal and colleagues [16–18]. In dealing with the daily incidence of new diagnosed COVID-19 cases, Poisson regression was used. The basic regression equation is defined as follows:

$$Y_t = \exp(\beta_0 + \beta_1 T + \beta_2 X_t + \beta_3 T X_t + \varepsilon_i)$$

where $Y_t$ is the outcome (COVID-19 daily incident cases) at time $t$, $T$ is the number of days elapsed since the beginning of the study, $X_t$ is a dummy variable indicating the pre-intervention period (coded 0) or the post-intervention period (coded 1), $\beta_0$ represents the baseline level at $T = 0$, $\beta_1$ is interpreted as the change in the outcome associated with a time unit increase (representing the underlying pre-intervention trend), $\beta_2$ is the level change following the intervention, and $\beta_3$ indicates the slope change following the intervention (the interaction term between time and intervention: $TX_t$). Hypothesising that the intervention would induce a change in the gradient of trend, yet not an immediate change in the level of the outcome, we omitted the dummy variable $\beta_2 X_t$ from our models. Moreover, it is reasonable to assume that the intervention would not exert its effects immediately at the cut-off; hence, we opted for a lagged impact model with slope change only [17]. Bearing in mind the average incubation period of the virus [19–21], we tested each model with a cut-off established at six days from the intervention date.

We tested three main models with the same specification, while differing in age of the population included in the analysis. Firstly, we examined the effect of the intervention in the overall ATS territory without Bollate (192 municipalities) exclusively in children aged between 3–11 years old, that is the population directly affected by the school closures (nursery schools for

children younger than 3 years old were not affected by the intervention, whilst higher grades were already put in blended distance learning before the intervention). Secondly, the model was tested in the same area in individuals aged between 12–19 years old, which were subject to a discontinuous alternation between proximate and distance learning before the intervention, according to local epidemic trends. Thirdly, the model was tested in individuals aged 20 years or more to examine the indirect effects of school closures on the spread of contagion in the overall population who had already finished their secondary education. As the daily number of incident cases is influenced by the number of swab tests performed, which is in turn influenced by the day of the week (a lower amount of tests is systematically performed on the weekend, especially on Sunday), the day of the week (to account for seasonality) and the daily number of swab tests (to account for time-varying confounding factors) were fitted as covariates. The models were also adjusted for the scenario of reference ('yellow', 'orange', 'strengthen orange', 'red') to take into account background restrictions in observations away from the cut-off. To ensure that the effects detected were independent of the overall local epidemic trend, the models were also adjusted for the daily number of incident cases and the positivity rate in the whole country. The reference population (log transformed) was included in the Poisson models as an offset variable in order to transform the count data into rates. The models were also tested without covariates but with Fourier terms to adjust for seasonality [22], and with a cut-off of 14 days from the intervention, corresponding to the maximum incubation period in the normal range [23]. Overdispersion was allowed by adding a scale parameter, thereby permitting the variance to be proportional rather than equal to the mean. Autocorrelation was checked by plotting the residuals against time and by examining the autocorrelation and partial autocorrelation functions. The model parameters were tested by z-tests and the statistical significance was set by P-values $< 0·05$ (2-sided); the results were expressed in terms of incidence rate ratios (IRRs). Additionally, we computed Kaplan-Meier survival curves among tested negative subjects to compare the contagion dynamics between the three age groups during the same time interval, also performing Cox regression and Wilcoxon tests to assess the equality of survival curves in the three age groups before and after the intervention. All the analyses were conducted using Stata version 17.

## Ethics statement

Ethical approval and consent to participate were not required, as this is an observational study based on data routinely collected by the ATS of Milan, a public body of the Regional Health Service–Lombardy Region. The ATS has among its institutional functions, established by the Lombardy Region legislation (R.L. 23/2015), the government of the care pathway at the individual level in the regional social and health care system, the evaluation of the services provided to, and the outcomes of, patients residing in the covered area. This study is also ethically compliant with the National Law (D.Lgs. 101/2018) and the "General Authorisation to Process Personal Data for Scientific Research Purposes" (nos. 8 and 9 of 2016, referred to in the Data Protection Authority action of December 13, 2018). Data were anonymized with a unique identifier in the different datasets before being used for the analyses.

## Results

Descriptive statistics concerning the distribution of cases, swab tests, and positivity rate in the two settings are reported in S1 Table in the S1 File. In the time frame considered (64 days, of which 32 pre- and 32 post-intervention), new 57,331 COVID-19 cases were detected out of 776,534 performed, leading to a 7.4% positive rate.

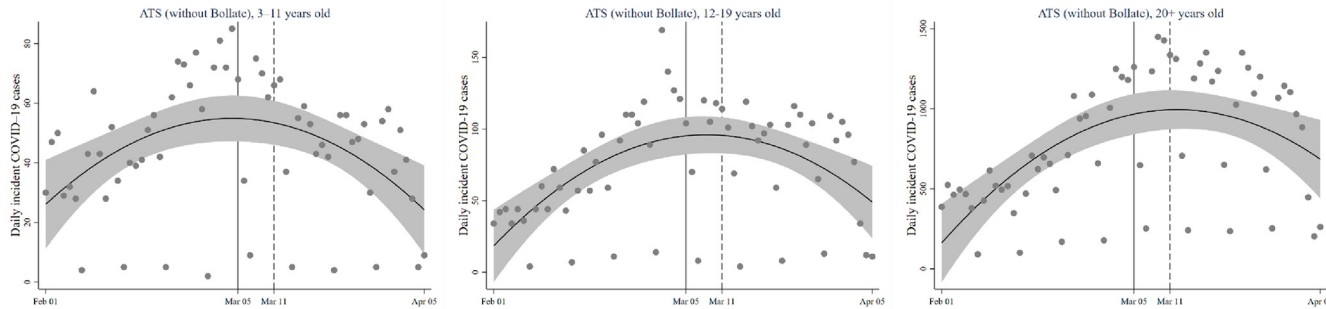

**Fig 1. Scatter plot and fractional polynomial line (degree = 2) with 95% confidence intervals of daily incident COVID-19 cases in the ATS territory (without Bollate) in individuals aged 3–11, 12–19, and 20+ years old or more.**

Fig 1 reports the raw distribution of daily COVID-19 cases before and after the school closure intervention in the three age groups. Together with the overall trend, the scatter plots highlight a noticeable degree of weekly seasonality, as the dots at the bottom of the graphs are related to the systematic reduced number of swab tests performed on Sundays. In all graphs, after a steady increase in the contagion curve, a decline after the intervention is visible. In the 3–11 and 12–19 age class, the decrease is located immediately after the cut-off, whereas in those aged 20 or above a more pronounced lag is present, which is consistent with the fact that the intervention exerted its effects in the first instance on those at whom it was directed (school-aged children), with a subsequent indirect effect on the remaining population.

Table 2 reports the results of the main models. The coefficient of interest is the interaction term ($TX_t$) between the dummy variable ($X_t$) expressing the cut-off and the continuous variable ($T$) indicating the time elapsed since the start of the study. This represents the change in

**Table 2. Segmented Poisson regression of daily incident COVID-19 cases before and after the school closure intervention (cut-off = six days).** ATS (without Bollate), individuals aged 3–11, 12–19, and 20+ years old or more.

| | 3–11 y.o. | | | 12–19 y.o. | | | 20+ y.o. | | |
|---|---|---|---|---|---|---|---|---|---|
| | IRR | P-value | [95% C.I.] | IRR | P-value | [95% C.I.] | IRR | P-value | [95% C.I.] |
| **Cut-off*time** | 0·96 | 0·004 | [0·94–0·99] | 0·96 | 0·005 | [0·94–0·99] | 0·97 | 0·000 | [0·96–0·98] |
| **Time** | 1·02 | 0·000 | [1·01–1·04] | 1·03 | 0·000 | [1·02–1·05] | 1·02 | 0·000 | [1·01–1·03] |
| **Scenario** | | | | | | | | | |
| *'yellow' (ref·)* | 1 | - | 1 | 1 | – | 1 | 1 | – | 1 |
| *'orange'* | 0·96 | 0·782 | [0·74–1·26] | 0·95 | 0·720 | [0·74–1·23] | 0·91 | 0·134 | [0·80–1·03] |
| *'strengthen orange'* | 0·71 | 0·042 | [0·51–0·99] | 0·68 | 0·014 | [0·50–0·93] | 0·82 | 0·005 | [0·71–0·94] |
| *''red'* | 0·65 | 0·033 | [0·44–0·97] | 0·71 | 0·060 | [0·49–1·01] | 0·83 | 0·028 | [0·70–0·98] |
| **Weekday** | | | | | | | | | |
| *Sunday (ref.)* | 1 | - | 1 | 1 | – | 1 | 1 | – | 1 |
| *Monday* | 10·62 | 0·000 | [5·98–18·87] | 6·52 | 0·000 | [3·70–11·47] | 1·49 | 0·019 | [1·07–2·08] |
| *Tuesday* | 10·09 | 0·000 | [6·01–16·93] | 5·60 | 0·000 | [3·35–9·38] | 2·16 | 0·000 | [1·66–2·80] |
| *Wednesday* | 8·64 | 0·000 | [5·38–13·88] | 5·44 | 0·000 | [3·36–8·80] | 2·06 | 0·000 | [1·62–2·62] |
| *Thursday* | 8·21 | 0·000 | [5·00–13·48] | 5·10 | 0·000 | [3·05–8·54] | 1·88 | 0·000 | [1·48–2·40] |
| *Friday* | 8·97 | 0·000 | [5·43–14·82] | 5·42 | 0·000 | [3·22–9·11] | 1·84 | 0·000 | [1·43–2·38] |
| *Saturday* | 6·25 | 0·000 | [3·91–9·99] | 4·06 | 0·000 | [2·56–6·45] | 1·89 | 0·000 | [1·60–2·23] |
| **Swab tests (ATS)** | 1·00 | 0·367 | [1·00–1·00] | 1·00 | 0·002 | [1·00–1·00] | 1·00 | 0·000 | [1·00–1·00] |
| **New cases (Italy)** | 1·00 | 0·309 | [1·00–1·00] | 1·00 | 0·613 | [1·00–1·00] | 1·00 | 0·356 | [1·00–1·00] |
| **Positivity rate (Italy)** | 0·94 | 0·278 | [0·85–1·05] | 0·94 | 0·216 | [0·85–1·04] | 1·04 | 0·114 | [0·99–1·10] |

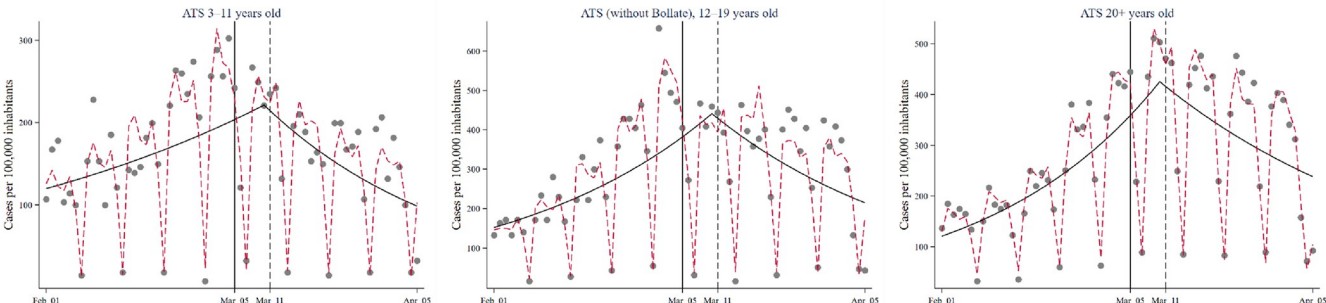

**Fig 2. Interrupted time series of daily incident COVID-19 cases in the ATS territory (without Bollate) in individuals aged 3–11, 12–19, and 20+ years old or more.** Solid line: predicted trend based on the covariate-adjusted segmented Poisson regression model. Dashed line: unadjusted trend.

the slope following the intervention. Specifically, the change in the slope is statistically significant in the models for all the three age groups. On average, the intervention produced a daily reduction of the risk of contagion by 4% in those aged 3–11 and 12–19 (IRR = 0·96) and by 3% in those aged 20 or more (IRR = 0.97). Results are adjusted for seasonality and time-varying confounders, meaning that the effects detected are not attributable to concurrent factors such as the underlying scenario applied, the epidemic trend in the whole country, or the number of swab tests performed locally or nationally. However, these controls cannot exclude the role of other unknown or unconsidered factors that could have influenced the trends (not available in data).

Fig 2 displays the results graphically, showing the predicted values based on the Poisson segmented regression models. The dashed lines report the predictions from the adjusted models presented in Table 2, from which it could be difficult to grasp the change in the trend due to the adjustment for seasonality. Hence, the predictions from unadjusted models have been overlaid on the plot (continuous lines) to facilitate comprehension. As is visible, for all the age groups there is a linear growth in the daily incidence up to the cut-off point set at six days from the intervention, in relation to which a reversal of the pattern is present, with the incidence starting to decrease linearly. Diagnostics, seasonality-adjusted models with Fourier terms, and models with a cut-off set at 14 days from the interventions, are presented in the Supplementary Material (S1–S15 Figs; S2 Table in S1 File).

Finally, the epidemiological curves in the three age groups have been compared using Kaplan-Meier survival functions. The analysis does not include censored cases (subjects who tested negative and/or subjects not tested), so it cannot be used to evaluate the risk of contagion. Rather, it allows for examining the differences in the contagion dynamics between the three age groups during the same time interval, which were found to be statistically significant (Fig 3). These, together with the Cox regression-based test for equality of survival curves [24] (Table 3), provided evidence that the contagion in the 3–11 age group was spreading more rapidly than in the other two age groups before the school closures—with a relative hazard of 1·13, compared to 1·05 for the 12–19 and 0·99 for the 20+ age group—but it became similar to other groups after the intervention, without statistically significant differences. Wilcoxon tests (P-value < 0.001) also confirmed the statistical significance of the differences between the Kaplan-Meier curves.

## Discussion

After the school closures, a considerable decline in daily COVID-19 incidence was visible. This was rather immediate for those directly addressed by the intervention (children 3–11 y.

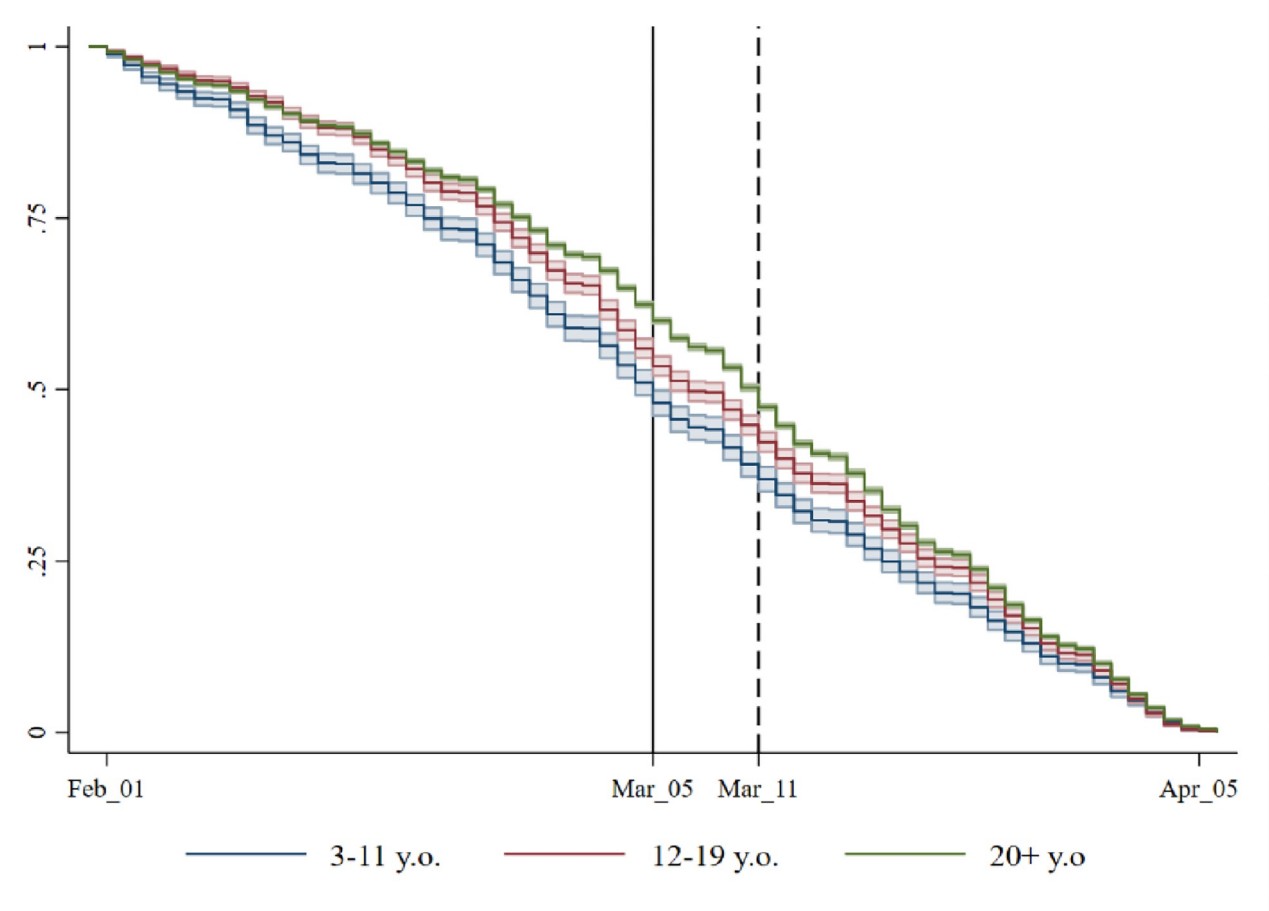

**Fig 3. Kaplan-Meier survival curves of daily incident COVID-19 cases in the three age groups.**

o.), while it was slightly delayed for the adult population (20+ y.o.), with a barely slower decrease, and subjects aged between 12 and 19 years in an intermediate situation. Being aware that the effects detected are not entirely ascribable to the intervention examined due to a portion of unadjustable confounders, it is nonetheless clear that school closure contributed substantially to mitigate the spread of the Alpha variant among the population. This evidence is supported by the following: (i) the presence of a visible discontinuity in the contagion curve right after the intervention, first occurring in school-aged children and only subsequently in the wider population, with the intermediate age group (12–19 y.o.) who experienced an

**Table 3. Cox regression-based test for equality of survival curves in the three age groups before and after the intervention (cut-off = six days).**

|  | Pre-intervention | | | Post-intervention | | |
|---|---|---|---|---|---|---|
|  | Likelihood Ratio = 36·05 | | | Likelihood Ratio = 0·010 | | |
|  | P-value = 0·000 | | | P-value = 0·996 | | |
|  | Events observed | Events expected | Relative hazard | Events observed | Events expected | Relative hazard |
| *3–11 y.o.* | 1,741 | 1,542 | 1·13 | 1,118 | 1,118 | 1·00 |
| *12–19 y.o.* | 2,611 | 2,478 | 1·05 | 2,122 | 2,118 | 1·00 |
| *20+ y.o.* | 24,724 | 25,056 | 0·99 | 25,015 | 25,019 | 1·00 |
| Total | 29,076 | 29,076 | 1·00 | 28,255 | 28,255 | 1·00 |

alternation of proximate and distance learning being in an intermediate situation. In relation to this, further evidence was provided by estimating survival curves from individual data, which compared the contagion curves between the three age groups. Specifically, the analysis of relative hazards shows that the raw contagion was spreading more rapidly in the younger group than in other age groups before the intervention and started to slow down earlier after the school closures; (ii) the triviality of concurrent restrictive measures (the prohibition against using playgrounds in public parks, reaching second homes, visiting friends and relatives without a 'proven' necessity; the closure of museums, bars, and restaurants; the prohibition against moving outside the municipality of residence), which—as discussed above—introduced minor changes and/or were subject to exceptions, resulting in having concurrent restrictive measures which were less stringent than it might appear.

More than the magnitude of the IRR coefficients emerged from the main models, which were not significantly different between the age groups, it is the temporal order of the reduction in infections between them to suggest a relevant contribution of the school closure intervention in lowering the epidemic curve. Indeed, whilst children have been generally underrepresented among COVID-19 cases in the first year of the pandemic and schools were not identified as a relevant nidus of transmission, the school closure intervention in correspondence of the novel variant highly contagious also among kids led to a lowering of the epidemic curve first in school-aged children directly affected by the intervention, and only subsequently in the adult population, which presumably indirectly benefitted from the reduction in infections among children. The temporal order of the events should also exclude the possibility of other causation mechanisms, such as a decrease in contagion as a consequence of parents forced to stay at home to take care to their children, as otherwise the variations in the age-stratified epidemic curves should be concomitant. Although new variants with greater transmissibility did not seem to give rise to more severe outcomes in children [25, 26], it is nonetheless important to highlight that children could have acted as spreader among the oldest and most fragile population, especially in a context where the national vaccination campaign was at its beginning. On the other hand, given the high social costs imposed by such an intervention in terms of parents' caregiving burden, children's education, psychosocial development and mental well-being [27, 28], it is clear that school closure should be imposed only basing on proven evidence of increasing and uncontrollable spread of infection among the school-aged population.

As limitations, we warn that (i) information on SARS-CoV-2-positive individuals was based on the swab test date—and not on the exact onset of the disease—implying a minimum amount of delay intrinsic to our data; (ii) the choice of the cut-off is necessarily somehow arbitrary, as we tested two different thresholds relying on the average and maximum incubation period reported by the literature, but in the lack of information concerning the specific case investigated this may be a source of bias; (iii) though concurrent restrictive measures presumably played a minor role compared to school closures, it is nonetheless important to stress that the IRR detected are likely to overestimate the real impact of the intervention on the decrease of infections, given that it was not possible to take into account the effects of unknown or unmeasured concurrent factors influencing the local epidemic trend.

School closures played a crucial role in reducing SARS-CoV-2 infections during the outbreak of the Alpha variant in the Milan metropolitan area, consequently limiting hospitalisations and casualties. Nonetheless, our study faced the issue from a purely epidemiological standpoint, without addressing the fundamental dispute concerning whether the above-mentioned related social costs exceeded the benefits of intervention, which goes beyond the scope of this study. Despite this, the claim that 'schools are safe from COVID-19', which was widely spread among the Italian media, appeared not to be supported by the empirical evidence.

## Supporting information

**S1 File. Supporting information–contains all the supporting tables and figures.**
(DOCX)

**S1 Dataset. Minimal dataset.**
(CSV)

## Author Contributions

**Conceptualization:** David Consolazio, Simone Sarti, Marco Terraneo, Corrado Celata, Antonio Giampiero Russo.

**Data curation:** Antonio Giampiero Russo.

**Formal analysis:** David Consolazio.

**Methodology:** David Consolazio, Simone Sarti, Marco Terraneo.

**Project administration:** Antonio Giampiero Russo.

**Supervision:** Antonio Giampiero Russo.

**Writing – original draft:** David Consolazio, Simone Sarti, Marco Terraneo.

**Writing – review & editing:** David Consolazio, Simone Sarti, Marco Terraneo, Corrado Celata, Antonio Giampiero Russo.

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
