## [Decision Letter · Decision Letter 0]

20 Apr 2022

PONE-D-21-36674

The impact of school closure during the third wave of the COVID-19 pandemic in Italy: evidence from the Milan area

PLOS ONE

Dear Dr. Russo,

Thank you for submitting your manuscript to PLOS ONE. After careful consideration, we feel that it has merit but does not fully meet PLOS ONE’s publication criteria as it currently stands. Therefore, we invite you to submit a revised version of the manuscript that addresses the points raised during the review process.

We look forward to receiving your revised manuscript.

Kind regards,

Hao Xue

Academic Editor

PLOS ONE

3. Please ensure that you include a title page within your main document. We do appreciate that you have a title page document uploaded as a separate file, however, as per our author guidelines (http://journals.plos.org/plosone/s/submission-guidelines#loc-title-page) we do require this to be part of the manuscript file itself and not uploaded separately.

Could you therefore please include the title page into the beginning of your manuscript file itself, listing all authors and affiliations.’

5. PLOS requires an ORCID iD for the corresponding author in Editorial Manager on papers submitted after December 6th, 2016. Please ensure that you have an ORCID iD and that it is validated in Editorial Manager. To do this, go to ‘Update my Information’ (in the upper left-hand corner of the main menu), and click on the Fetch/Validate link next to the ORCID field. This will take you to the ORCID site and allow you to create a new iD or authenticate a pre-existing iD in Editorial Manager. Please see the following video for instructions on linking an ORCID iD to your Editorial Manager account: https://www.youtube.com/watch?v=_xcclfuvtx.

6. Please amend your authorship list in your manuscript file to include author’s list.

Reviewers' comments:

Reviewer's Responses to Questions

**Comments to the Author**

1. Is the manuscript technically sound, and do the data support the conclusions?

Reviewer #1: Yes

Reviewer #2: Yes

Reviewer #3: No

2. Has the statistical analysis been performed appropriately and rigorously? 

Reviewer #1: Yes

Reviewer #2: I Don't Know

Reviewer #3: No

3. Have the authors made all data underlying the findings in their manuscript fully available?

Reviewer #1: Yes

Reviewer #2: Yes

Reviewer #3: No

4. Is the manuscript presented in an intelligible fashion and written in standard English?

Reviewer #1: Yes

Reviewer #2: Yes

Reviewer #3: Yes

5. Review Comments to the Author

Reviewer #1: I would like to congratulate the authors for the study as it would help to take action by the concerned authorities in preventing the spread of the COVID-19 and in pandemics in future too. Though there could be many confounding factors and bias in the decreasing hospitalization and casualties that the authors had mentioned in the limitation. There are also other school of thoughts on the risk vs benefit of social costs and other concerns of school closure.

Overall the study is appreciable in terms of analysis, interpretation and publucatiton.

Reviewer #2: Report for “The impact of school closure during the third wave of the COVID-19 pandemic in Italy: evidence from the Milan area.”

This study evaluates the impact of school closure policy on reducing the spread of COVID-19 pandemic. The authors use an interrupted time series design by comparing right before and right after the implementation of the policy.

Major comments:

1. As acknowledged by the authors, it is very hard to control for other concurrent policies or events in this design. Is it possible to find a region that is close to the Milan area but didn’t implement the policy on March 5th to serve as a control group? We can show that the infection in the control region didn’t change much after March 5th. By doing so, we can at least control for factors such as national level policy variation, change in climate and people’s awareness.

2. Another way to clarify the mechanism is to investigate whether the effect of the policy is more significant in regions with more schools. If the effect is really driven by school closure, I think this should be the case.

3. School closure may force some parents to stay at home and take care of their children, which will also reducing transmission. This can also become a mechanism through which school closure affect transmission.

Reviewer #3: Referee report for “The impact of school closure during the third wave of the COVID-19 pandemic in Italy: evidence from the Milan area”

Summary: The authors attempt to address the important question of the effect of school closures on the spread of COVID-19, focusing on the Lombardy area in Italy during an outbreak in Feb – Apr 2021. The authors concentrate on children ages 3 to 11, whose schools were shut down on March 5th until April 5th. Older students’ schools had already been shut down for a couple of weeks by the time the 3-11 year-olds’ schools were closed. The authors estimate Poisson regression models for COVID cases for 3-11 year-olds and 20+ year-olds separately (also 12-19 in an appendix), pursuing an interrupted time-series research design, allowing for a slope break 6 days after the schools closed. They find that after the schools shut down, case counts decreased by about 4% for 3-11 age group and 3% for the 20+ group. While the authors do not test for differences between estimates in these two groups, it seems quite clear from their results and plots that these 4% and 3% estimates are not statistically distinguishable. I would argue they represent the same effect. The authors then conclude that, “it is nonetheless clear that school closure contributed substantially to mitigate the spread of the Alpha variant among the population” (p. 15).

Comments

1) I think the authors address an important question and they have some nice data on covid cases by age in the Milan, Italy area.

2) That said, I do not think the authors employ a credible research strategy in this case with their interrupted time series. This is an obsolete strategy and should be replaced with more modern approaches. In the authors’ case, a difference-in-differences (DD)/event study approach would be a modern and credible research design that I think could be feasible for the authors given their data. The problem with the interrupted time series approach is the authors can’t account for other factors that could be correlated with time. Other COVID-19 precautions implemented at the same time as the school closures are one of these other factors. The authors discuss these and clam they are likely to have little effect (without evidence). They might have a point, but that doesn’t mean that we shouldn’t use available methods to be more confident, rather than just having to take the authors’ words for it. More important, though, is the fact that populations respond strongly to COVID-19 out-breaks on their own, without the government having to force them to change their behavior. We have seen this in many ways during the COVID-19 pandemic. Thus, while the government responds to the covid outbreak, so do the people in changing their behavior. This change in behavior could be the driving factor for the fall in cases (or a contributor). The authors’ interrupted time series does not allow for any way to control for these factors.

3) The authors would be much better trying to provide a control group for comparison, as this group would control for the behavior of the population, the other government restrictions, and any other omitted factors. Looking at their data, it seems like the authors could have used 12-19 year-olds as such a group, or even the 20+ year-olds, though 12-19 is the better group as most of that group also attends school. The authors note in Table 1 that with entry into the “yellow zone” on February 1, schools at the second level of secondary school and above (i.e., 12 year-olds and up) had their schools closed. So why not compare cases among this group to those of 3-11 year-olds who schools weren’t closed? The 3-11 age group would control for population response to covid and also for the other government restriction that were imposed. The difference in cases between the groups could then more credibly be called the effect of school shut-down. They could even use variation from when the younger kids’ schools were shut down as part of this approach, as well. Alternatively, the authors could use the 20+ year-olds as a control group. This is not as strong a control group since these people don’t attend school, but it is far better than the approach the author currently use where there is no control for population response at all.

4) Looking at the authors’ data, we can informally perform a sort of DD approach by comparing how case rates changed over time across these three groups. The results are presented in Table 2 and in Table S2 (in the appendix). While the authors don’t formally test for differences, it is clear that the estimates of spread in the authors’ pre-period (estimate for variable “time”) are almost quantitatively identical. Moreover, during the post-period, they are still almost quantitatively the same. Without a doubt, across the groups these estimates would not be distinguishable statistically. Thus, in my view the authors’ data is suggestive of their being no effect of the school shut-downs. That said, if the authors had performed a more formal DD style analysis, maybe they would have been able to show a difference. But at this point, I see their analysis as suggesting no effect.

5) Given the above, I think the authors’ conclusions are far too strong, and are not supported by the data they present. In particular, I think the following statement made by the authors is unsupported by their data: “Despite this, the claim that ‘schools are safe from COVID-19’, which was widely spread among the Italian media, appeared not to be supported by the empirical evidence.” (p. 16) (After reviewing their evidence, I am *more* inclined to agree that there was no impact of the school closures and that schools are safe). Additionally, the conclusion I noted in the summary, “it is nonetheless clear that school closure contributed substantially to mitigate the spread of the Alpha variant among the population” (p. 15), is also not supported by the authors’ data. I would suggest the authors remove these statements, and any other similar ones.

Minor comments

a) Would have been helpful if the draft had included page numbers. My page numbers are the page in the PDF file I was provided.

b) The authors’ regression equation should have an exponential function in it given that it is supposed to be a poisson model, and should also have an error term.

c) P. 13: “Bearing in mind the average incubation period of the virus, we tested each model with a cut-off established at six days from the intervention date.” The authors could be clearer about what they mean by this. For a while as I was reading I thought it meant the authors lagged the data by 6 days.

d) Figure 2: I really don’t see a point to including the red-dashed lines. They just cause confusion. Plus I don’t think the authors ever really explain what it is.

e) Was school actually closed on March 5? It is not clear from the manuscript. That was a Friday, and though I am not sure if Italy follows the same system as the USA, but if schools were closed for Saturday and Sunday, maybe the authors should not consider the school closures to be implemented until March 8, the Monday in which might have been closed for the first time?

f) P. 14: “Results are adjusted for seasonality and time-varying confounders, meaning that the effects detected are not attributable to concurrent factors such as the underlying scenario applied, the epidemic trend in the whole country, or the number of swab tests performed locally or nationally.” – It is helpful to include these controls, but I think this statement is entirely too strong. One could argue that these controls are not sufficient to fully control for these factors (and could do so quite effectively, I might add), so I would not say these factors could not have *any* effect on the authors’ results. I would say that they include these controls which help address concerns that these factors could be the true causes of the authors’ estimates.

g) It is odd the way the authors at times reference the 12-19 age group yet don’t include their results in the paper (only an appendix).

---

## [Author Response · Author response to Decision Letter 0]

25 May 2022

Dear Reviewers, 

We thank you for your comments and suggestions, which we believed have helped us in reaching a higher quality output. We provide point-by-point answers to each comment in the file attached.

Overall, we made efforts to make our findings more comprehensible to the readers, as we believe that in the first version we did not stress enough the importance of the temporal dimension in suggesting the relevance of the school closure intervention. It is not so much the magnitude of the coefficients that convinced us about the effectiveness of the intervention, but rather the fact that the contagion curve started to decrease first in those directly affected by the intervention, and only subsequently in the adult population, which benefited from the reduction in contagion in school-aged children, only marginally affected by the virus before the novel “British” strain. 

Despite we included several time-varying confounders, we agree with the reviewers that this might not be enough to completely separate the effect of the intervention from other unmeasured concomitant variations, and then that the decrease in the contagion curve could be attributable to factors other than the school closure. However, the temporal order of contagion decrease in different age groups is a very relevant point in favor of the effectiveness of the intervention, as we found no other explanation for this robust and meaningful evidence. 

Hence, we deeply revised to manuscript to highlight this issue. 

Yours sincerely, 

The authors

---

## [Editor Report · Decision Letter 1]

30 Jun 2022

The impact of school closure intervention during the third wave of the COVID-19 pandemic in Italy: evidence from the Milan area

PONE-D-21-36674R1

Dear Dr. Russo,

We’re pleased to inform you that your manuscript has been judged scientifically suitable for publication and will be formally accepted for publication once it meets all outstanding technical requirements.

Kind regards,

Hao Xue

Academic Editor

PLOS ONE

---

## [Editor Report · Acceptance letter]

4 Jul 2022

PONE-D-21-36674R1 

The impact of school closure intervention during the third wave of the COVID-19 pandemic in Italy: evidence from the Milan area 

Dear Dr. Russo:

I'm pleased to inform you that your manuscript has been deemed suitable for publication in PLOS ONE. Congratulations! Your manuscript is now with our production department. 

Kind regards, 

on behalf of

Dr. Hao Xue 

Academic Editor

PLOS ONE